# The bone bridge significantly affects the decrease in bone mineral density measured with quantitative computed tomography in ankylosing spondylitis

So Yun Lee[1], Ran Song[1], Hyung In Yang[1], Sang Wan Chung[1], Yeon-Ah Lee[1], Seung-Jae Hong[1], Seong Jong Yun[2], Sang-Hoon Lee[1]*

1 Department of Rheumatology, College of Medicine, Kyung Hee University, Seoul, Korea, 2 Department of Radiology, College of Medicine, Kyung Hee University, Seoul, Korea

* boltaguni@gmail.com

## Abstract

**Data Availability Statement:** All relevant data are within the paper.

### Introduction and objective

Ankylosing spondylitis (AS) has characteristics of spinal bone bridge and fusion. Although BMD reduction in AS may be presumed to be due to spinal inflammation, this study was designed to confirm whether immobilization of the spine due to syndesmophytes is related to BMD reduction, as immobilization itself is a risk factor for BMD reduction.

### Methods

Among male patients diagnosed with AS according to the modified New York criteria, those who underwent bone density tests with quantitative computed tomography (QCT) were retrospectively analyzed through a chart review. The correlation between the presence or absence of bone bridges for each vertebral body level of the L spine confirmed with radiography and BMD confirmed with QCT was analyzed.

### Results

A total of 47 male patients with AS were enrolled. The mean patient age was 46.8 ± 8.2 years, and the mean disease duration was 7.9 ± 6.4 years. The trabecular BMD of the lumbar spine (L1-L4) ranged from 23.1 to 158.45 mg/cm$^3$ (mean 102.2 ± 37 mg/cm$^3$), as measured with QCT. The lumbar BMD measurements showed that 30 patients (63.8%) had osteopenia or osteoporosis. Bone bridge formation showed a negative correlation with BMD. Low BMD was significantly correlated with bone bridge in the vertebral body ($p < 0.05$). Positive correlations were observed between bone bridge score and BASMI flexion score, whereas significant negative correlations were found between BMD and BASMI flexion score ($p < 0.05$).

**Funding:** The author(s) received no specific funding for this work.

**Competing interests:** The authors have declared that no competing interests exist.

## Conclusion

Decreased mobility of the vertebrae due to bone bridge formation affects the decrease in BMD in patients with AS.

## Introduction

Ankylosing spondylitis (AS) is a chronic inflammatory disease that primarily affects the sacro-iliac joints and the axial spine, and it also affects the peripheral joints. Two of the bone remodeling process occurs in AS. Syndesmophyte is produced through bone formation in the spine, and the process of bone loss increases the risk of osteoporosis and fractures [1].

In most cases of AS patients, the burden of disease is due to a combination of structural bone damage and inflammation [2]. Radiographic progression of the spine is indicated by the formation of syndesmophyte leading to bridging of the intervertebral spaces. Structural damage acts on patients by causing the loss of disability and permanent loss of function [3].

Chronic inflammation of the spine leads to the formation of new bones in the axial spine and vertebral spaces, as well as bone resorption, leading to an increase in osteoporosis in AS [4, 5]. It has been demonstrated that the general bone loss may be due to systemic inflammation and disease activity [5, 6]. Because the disease activity of AS affects the rate of bone loss, osteoporosis is thought to be a manifestation of the disease itself and not a comorbid condition [7]. In previous studies it has been proven to have a higher prevalence of osteoporosis and significantly lower bone mineral density (BMD) in AS patients than in sex- and age- matched controls [8, 9].

In immobilized patients, low BMD and abnormal bone metabolism have been reported [10, 11]. Loss of spinal mobility is a major feature of AS, and it has been hypothesized that BMD reduction is related to immobility due to ankylosis in patients with AS [12]. In a previous study, Baraliakos et al. showed that patients with bridging syndesmophytes had significantly higher levels of procollagen type 1 N-terminal peptide, a bone formation marker, and serum collagen-telopeptide, a bone resorption marker. In addition, they found that BMD was lower in patients with syndesmophytes [13].

Dual-energy x-ray absorptiometry (DXA) is the usual method of measuring BMD [14]. However, in patients with AS, it may be difficult to assess osteoporosis with lumbar spine BMD measured with DXA. Even in precence of osteoporosis due to the new bone formation characteristic of AS, the value can be normal or high, leading to an overestimation of the total BMD [15]. Spinal hyperostosis of AS often occurs around the zygapophyseal joints, vertebral endplates, and annulus fibrosus of the discs [16].

Quantitative computed tomography (QCT) is a technology that utilizes a calibration standard imaged with a patient in a standard CT scanner to allow the calibration of gray-scale CT image values in terms of BMD. With QCT, volumetric trabecular bone density can be measured without overlap of cortical bone and other tissues, and results are expressed in milligrams per cubic centimeter of calcium hydroxyapatite [17]. Li et al. [17] demonstrated that QCT can avoid overestimating BMD by DXA associated with spinal degeneration and sclerosis lesions. Therefore, QCT may be more sensitive than DXA in detecting osteoporosis in patients with AS.

We planned this study to determine whether the formation of a bone bridge correlates with low BMD measured with QCT in patients with AS.

## Materials and methods

### Patients

Patients aged > 18 years diagnosed with AS who visited the Department of Rheumatology of Kyung Hee University Hospital, Gangdong, Korea, over a period of 9 years (2010–2018) were retrospectively reviewed. All patients met the 1984 modified New York criteria for definite AS, and a complete set of radiographs were available for each patient [18]. A total of 47 male patients who underwent QCT for BMD measurement were enrolled. Female patients were excluded to rule out the effects of osteoporosis from a postmenopausal status.

### Clinical data

In this retrospective, cross-sectional study, disease-related data and disease activity scores were collected at baseline. All medical records were obtained through a chart review. Clinical data included disease duration, and the presence of human leukocyte antigen-B27, uveitis, and peripheral arthritis. The Bath Ankylosing Spondylitis Metrology Index (BASMI) score [19] was also recorded for spinal mobility, and the erythrocyte sedimentation rate (ESR), C-reactive protein (CRP) level, and alkaline phosphatase (ALP) level were obtained. The use of medications such as non-steroidal anti-inflammatory drugs (NSAIDs), steroid, biologics, calcium, and vitamin D was recorded.

### Ethics statement

The Institutional Review Board (IRB) approval for the study was received, Informed consent was waived by the board. Retrospective data collection was approved by the IRB of Kyung Hee University Hospital, Gangdong (IRB No. 2019-08-012). After IRB approval, we accessed the record for 2 months from August 2019 to obtain the retrospective data used in our study.

### Radiographic scoring for ankylosis

An experienced two rheumatologist and one radiologist independently scored the radiographs for the presence or absence of a bone bridge for each intervertebral disc space (IDS) from T12–L1 to L4–L5. Lumbar spine radiography was used to evaluate the anterior and lateral sides of the vertebral body. The number of bone bridges was verified in the upper and lower bone bridge in one vertebra, in the anterior and lateral sides, and was scored up to a maximum of 6 points based on radiographic findings. The score was 0 if no syndesmophyte or < 90% syndesmophyte of the IDS was present and 1 if the syndesmophyte bridges the IDS or ≥ 90% syndesmophyte of the IDS was present. The schematic presentation and scoring are presented in Fig 1.

### Spinal mobility

We investigated the relationship between lumbar syndesmophytes and physical function in all patients. To assess physical function and lumbar spine mobility, the BASMI lumbar flexion score was obtained in patients with ankyloses [20]. The lumbar flexion range was measured using the modified Schober index, whereas the lateral flexion of the lumbar spine was bilaterally measured and the mean of the right and left flexion values was accepted as a single value. The scores were in the range of 0–10, with higher scores indicating more severe disease involvement.

A.

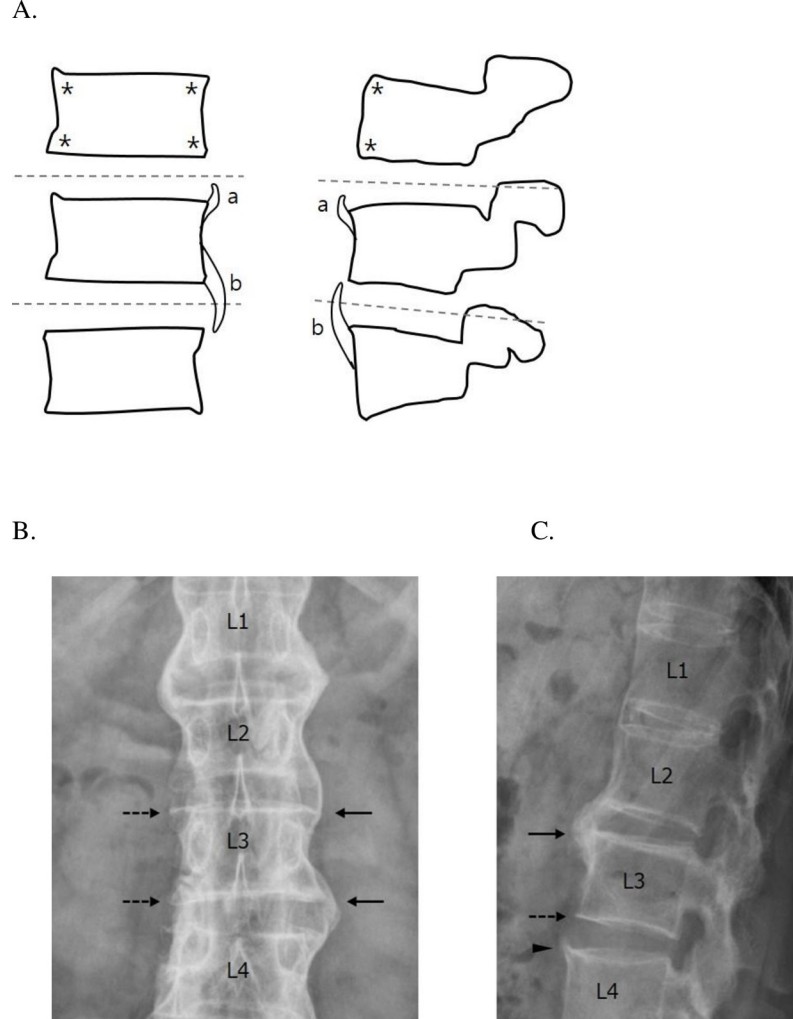

**Fig 1. Schematic presentation for bone bridge scoring.** (A) Schematic coronal and sagittal views of vertebrae with syndesmophytes. The gray, dotted line indicates the middle of the intervertebral disc space (IDS). Scoring was performed at the four corners of the coronal view and at the two corners of the anterior side of the sagittal view per vertebra (asterisk) with 0 or 1 point. A score of 0 is given for no syndesmophyte or < 90% syndesmophyte (a), and a score of 1 is given for ≥ 90% syndesmophyte of the IDS (b). (B) Anteroposterior radiographic image of the lumbar spine. At L3, a bridging syndesmophyte or ankylosis is seen (black arrow). A score of 1 is given for the upper left corner of L3 and the lower left corner of L3. No syndesmophytes are present at the right corner of L3 (dashed arrow), which is given a score of 0. (C) Lateral radiographic image of the lumbar spine. At the anterior side of L4, a syndesmophyte is growing from the upper corner (arrowhead) but does not reach ≥ 90% of the IDS, which is given a score of 0. At the anterior side of the vertebrae units L2–L3 (black arrow), a bridging syndesmophyte is seen (black arrow), and a score of 1 is given for the anterior upper corner of L3 and the anterior lower corner of L2. No syndesmophytes are present at the anterior lower corner of L3 (dashed arrow), which is given a score of 0.

## QCT analysis

QCT measurements were obtained with a Brilliance 64- slice CT scanner (Philips, Israel) with a solid Mindways QCT phantom (Mindways Software Inc., Austin, TX, USA). Patients were scanned at 120 kVp, 150 mA s, with exposure based on height and weight. Contiguous series of CT images were acquired with a 2.5-mm slice thickness and 2.5-mm spacing between images. Vertebrae from L1 to L4 were scanned in the supine position. Images were analyzed

using the Mindways software. Elliptical regions of interest were automatically placed in the midplane of four vertebral bodies (L1–L4) in the trabecular bone, avoiding the cortical bone of the vertebrae. Fractured vertebrae were excluded from measurement.

### BMD assessment

Average BMD is expressed in milligrams per cubic centimeter of calcium hydroxyapatite. For the BMD of the spinal trabecular bone, thresholds of 120 mg/cm$^3$ for osteopenia (equivalent to a DXA T-score of −1.0 standard deviation [SD]) and 80 mg/cm$^3$ for osteoporosis (equivalent to a DXA T-score of −2.5 SD) were suggested by the International Society for Clinical Densitometry in 2007 [21] and by the American College of Radiology in 2008 [22].

### Statistical analysis

Continuous data are expressed as mean ± SD, and categorical data are expressed as percentages. Clinical variables were compared using independent t tests, and the chi-squared test was used to compare categorical variables between patients with normal BMD and those with low BMD. A $p$-value of $\leq 0.05$ was considered statistically significant. Interobserver reliability was assessed using the intraclass correlation coefficients (ICCs). Statistical analysis was performed with IBM SPSS Statistics version 18 (SPSS corporation, USA). Power analysis was performed with PASS 16.0.9 software (NCSS, LLC, Kaysville, Utah, USA).

## Results

### Demographic features and initial laboratory data

A total of 47 patients were enrolled in the study, comprising 30 patients (63.8%) with osteopenia or osteoporosis and 17 patients (36.2%) with normal BMD as assessed with QCT. The demographic features and laboratory data of the enrolled patients are summarized in Table 1. The mean (SD) age was 46.8 (8.2) years, the mean (SD) disease duration was 7.9 (6.4) years, and the mean (SD) BMD measured with QCT was 102.2 (37) mg/cm$^3$. Two groups were compared according to the decrease in BMD. The two groups were divided based on 120 mg/cm$^3$, and the low BMD group included patients with osteopenia and osteoporosis. No significant differences were observed in inflammatory markers, ALP level, and treatment agents between the two groups. Eight patients (17%) were taking steroids at an average dose of < 5 mg/day based on prednisolone.

Among the variables, bone bridge and BASMI score showed significant differences between the two groups. Patients with low BMD showed higher bone bridge number and BASMI score ($p < 0.05$ and $p = 0.009$, respectively), as presented in Table 1.

### Correlation between bone bridge and QCT BMD

Table 2 shows the distribution of BMD and bone bridges per vertebral level in the lumbar spine. We analyzed the correlation between bone bridge number and BMD, and found significant negative correlations at all levels ($p < 0.05$). The ICCs for the bone bridge score was 0.944 for the lumbar spine and 0.914–0.954 for each vertebral body level of the L spine. As a result of post hoc analysis on the 47 patients used in this study, the power was 0.96 at ρ0 = -0.501, α = 0.05.

### BMD difference according to the existence of bone bridges

We analyzed the difference in BMD between two groups according to bone bridge status. Of the patients, 12 (25.5%) had no bone bridge and 35 (74.5%) had more than one bone bridge.

**Table 1. Baseline characteristics.**

| Variable | Total patients (N = 47) | Normal BMD (n = 17) | Low BMD (n = 30) | p Value |
|---|---|---|---|---|
| Age (years) | 46.8 ± 8.2 | 45.4 ± 7.7 | 47.6 ± 8.5 | 0.384 |
| Male sex, n (%) | 47 (100) | 17 (100) | 30 (100) | 1.000 |
| BMI (kg/m$^2$) | 25.1 ± 3.5 | 25.0 ± 3.1 | 25.1 ± 3.7 | 0.905 |
| Disease duration (years) | 7.9 ± 6.4 | 9.5 ± 6.1 | 7.0 ± 6.5 | 0.214 |
| HLA B27-positive, n (%) | 42 (89.4) | 15 (88.2) | 27 (90) | 0.333 |
| Bone bridge score (range 0–24) | 7.9 ± 8.1 | 2.6 ± 3.8 | 10.9 ± 8.5 | < 0.001 |
| BASMI score (range 0–10) | 3.3 ± 1.8 | 2.2 ± 1.5 | 3.7 ± 1.7 | 0.009 |
| BASMI lumbar flexion score (range 0–10) | 5.5 ± 1.8 | 4.2 ± 2.4 | 6.2 ± 2.8 | 0.029 |
| BASMI lumbar side flexion score (range 0–10) | 5.4 ± 2.8 | 3.1 ± 2.3 | 6.5 ± 2.9 | 0.001 |
| BMD (mg/cm$^3$) in QCT | 102.2 ± 37 | 141.7 ± 13.4 | 19.9 ± 25.4 | < 0.001 |
| ESR (mm/h) | 17.9 ± 21.4 | 16.6 ± 18.2 | 17.7 ± 23.4 | 0.932 |
| CRP (mg/dL) | 0.6 ± 1.0 | 0.7 ± 1.4 | 0.5 ± 0.8 | 0.610 |
| ALP (U/L) | 90.4 ± 24.4 | 87.9 ± 26.3 | 92.5 ± 23.5 | 0.429 |
| History of peripheral arthritis, n (%) | 13 (27.7) | 8 (47.1) | 5 (16.7) | 0.043 |
| History of uveitis, n (%) | 19 (40.4) | 6 (35.3) | 13 (43.3) | 0.599 |
| Patients on NSAIDs, n (%) | 45 (95.7) | 15 (88.2) | 30 (100) | 0.163 |
| Patients on PPI, n (%) | 5 (10.6) | 3 (17.6) | 2 (6.7) | 0.764 |
| Patients on TNF inhibitor, n (%) | 19 (40.4) | 9 (52.9) | 10 (33.3) | 0.314 |
| Patients on IL-17 inhibitor, n (%) | 1 (2.1) | 0 (0) | 1 (3.3) | 0.447 |
| Patients on steroid (recent 1 year), n (%) | 7 (14.9) | 1 (5.9) | 6 (20) | 0.379 |
| Total steroid usage (mg, prednisolone) | 1374.3 ± 778.6 | 1825 | 1299.2 ± 824.6 | 0.581 |
| Patients on calcium or vitamin D, n (%) | 1 (2.1) | 1 (5.9) | 0 (0) | 0.332 |

Data presented as n (%) or mean ± standard deviation.

BMI, body mass index; HLA-B27, human leukocyte antigen-B27; BASMI, Bath Ankylosing Spondylitis Metrology Index; BMD, bone mineral density; QCT, quantitative computed tomography; ESR, erythrocyte sedimentation rate; CRP, C-reactive protein; ALP, alkaline phosphatase; NSAIDs, nonsteroidal anti-inflammatory drugs; PPI, proton pump inhibitor; TNF, tumor necrosis factor; IL-17, interleukin-17.

The QCT BMD values were lower in the bone bridge group than in the no bone bridge group. Patients in the bone bridge group were significantly more osteopenic at L1–L4 ($p < 0.001$) (Table 3).

## Correlation between flexion score and QCT BMD

We analyzed the correlation between spinal mobility and BMD. Table 4 presents a comparative analysis of BMD and bone bridge score with the BASMI flexion score. Positive correlations

**Table 2. Correlation between bone bridge and QCT BMD.**

| Region | QCT BMD mean (mg/cm$^3$) | Bone bridge score | r* | p Value |
|---|---|---|---|---|
| L1 | 104.2 ± 41.7 | 2.6 ± 2.2 | -0.589 | < 0.001 |
| L2 | 102.3 ± 42.3 | 2.2 ± 2.3 | -0.652 | < 0.001 |
| L3 | 100.1 ± 40.8 | 1.7 ± 2.2 | -0.539 | < 0.001 |
| L4 | 102 ± 42 | 1.4 ± 2.0 | -0.501 | 0.001 |

Data presented as mean ± standard deviation.

* Pearson correlation coefficient.

BMD, bone mineral density; QCT, quantitative computed tomography.

**Table 3. BMD difference according to existence of bone bridge.**

| Region | QCT BMD mean (mg/cm$^3$) | | p Value |
|---|---|---|---|
| | Without bone bridge (n = 12) | With bone bridge (n = 35) | |
| L1 | 126.2 ± 37.2 | 96.7 ± 41 | < 0.001 |
| L2 | 126.2 ± 36.4 | 94.1 ± 41.5 | < 0.001 |
| L3 | 115.7 ± 39 | 94.4 ± 40.6 | < 0.001 |
| L4 | 127.4 ± 19.9 | 94.3 ± 44 | < 0.001 |
| L1–L4 | 123.9 ± 28.6 | 94.8 ± 37 | 0.017 |

Data presented as mean ± standard deviation.

BMD, bone mineral density; QCT, quantitative computed tomography.

were observed between bone bridge score and BASMI flexion score, whereas significant negative correlations were found between BMD and BASMI flexion score ($p < 0.05$).

## Discussion

This study in patients with AS analyzed the association between QCT BMD and bone bridge and flexion function. In vertebrae with bone bridges, the QCT BMD was statistically significantly lower, and flexion score and BMD were correlated with each other.

Several studies have measured the BMD of long-standing AS patients to investigate the prevalence of osteoporosis in AS. However, spinal BMD in patients with AS can be falsely elevated owing to ligamentous calcifications, sclerosis of the vertebral margins, and syndesmophyte formation [23]. This causes the problem of underestimation of the diagnosis of osteoporosis using DXA in patients with AS. To overcome these problems, we used QCT to evaluate BMD in this study.

Previous studies have shown an increased prevalence of osteoporosis in AS patients compared to sex and age-matched controls [8, 9]. In the general population, age-related osteoporosis was confirmed to occur in about 10% [24, 25]. In comparison, prospective data of AS showed osteoporosis prevalence rates between 21% and 25% [26, 27]. In a multicenter study in 204 patients with AS from western Sweden, osteoporosis was observed in 21%. Prospective data were also reported on 80 patients with AS in a Moroccan study, and osteoporosis was reported in 25% [27].

In this study, 63.8% of the enrolled patients showed a decrease in BMD, and osteoporosis was found in 29.8% (n = 14). This osteoporosis rate is higher than identified in several previously published studies in which BMD was measured using DXA. Lange et al. [28] reported an ideal method for measuring BMD in patients with AS, and their analysis found that the rate of osteoporosis was 9.2% using DXA and 30.3% using single- energy QCT analysis. This supports the results of our research.

**Table 4. Correlation of bone bridge and BMD with spinal mobility.**

| | BMD | | Bone bridge score | |
|---|---|---|---|---|
| | r* | p Value | r* | p Value |
| BASMI lumbar flexion score | -0.387 | 0.014 | 0.496 | 0.001 |
| BASMI lumbar side flexion score | -0.570 | < 0.001 | 0.483 | 0.002 |

* Pearson correlation coefficient.

BMD, bone mineral density; BASMI, Bath Ankylosing Spondylitis Metrology Index.

Bone loss and osteoporosis are well-recognized features of AS [29]. However, the specific cause of osteoporosis in AS is complex and potentially include several factors, and various theories about the cause have been proposed.

Previous prospective studies have shown that spine and hip BMD values are predominantly decreased in patients with active disease [30, 31]. In patients with early back pain, increased ESR and CRP, as well as bone inflammatory lesions on magnetic resonance imaging (MRI) were found to be determinants of low spine BMD [32]. In a 1-year prospective study, hip bone loss was found to be associated with elevated baseline CRP in a post hoc analysis and with sacroiliitis diagnosed with MRI [33]. All these data support the role of inflammation in bone loss in AS. In addition, several studies have shown significant correlations between markers of bone turnover, pro-inflammatory cytokines, and acute-phase reactants in AS [34–36]. This suggests that these systemic inflammatory mediators have a role in regulating bone turnover in AS. However, although we did not identify either bone turnover markers or pro-inflammatory cytokines, our study found no difference in ESR and CRP between patients with normal BMD and those with low BMD. It can be presumed that factors other than inflammation cause bone loss, which indirectly supports our theory.

Another early hypothesis suggested that immobility associated with pain, stiffness, and reduced physical activity from eventual ankylosis is the likely cause of bone loss in AS [12]. In previous studies, as osteoporosis was also present in patients with early disease who were devoid of any functional impairment, it was deemed likely that systemic factors also contribute to bone loss in AS [37]. However, it is important to consider that bone loss in patients with AS in the early stage occurs in combination with several factors. We found that the bone bridge was present and that higher scores mean lower BMD. If ankylosis occurs due to bone growth, then it may result in immobility and the decrease in physical activity may be an important factor in osteoporosis progression. Therefore, our results support this immobility theory.

Many widely used medications have now been shown to cause decreases or increases in BMD [38, 39]. Among them, NSAIDs and glucocorticoids for controlling inflammation in patients with AS, as well as proton pump inhibitors (PPIs), which are used together to reduce the adverse effects of drugs, are known to affect BMD. Glucocorticoids have a wide variety of direct and indirect effects on bone, as recently reviewed in detail by Henneicke et al. [40]. Carbone et al. [41] have shown that NSAID users had statistically significantly higher BMD values than nonusers at the cortical component of the lumbar spine. In addition, several large observational studies have suggested that PPI use is associated with a modest increase in osteoporotic fracture risk [42–45]. In our study, no difference in NSAID, glucocorticoid, and PPI use was found between the two groups. A positive effect on BMD was found in previous prospective open studies in patients with AS taking a tumor necrosis factor (TNF) inhibitor [39]. However, no significant differences in the use of TNF inhibitors were found between the two groups in our study.

This study had some limitations. First, other factors known to cause osteoporosis were not been sufficiently identified. There is a known association be among vitamin D, bone turnover markers, and BMD in AS. In addition, nonbiological factors, such as physiological, environmental, and lifestyle factors, are known to affect BMD. However, these variables were not measured in this study. To establish a direct link between bone bridge formation and BMD reduction, further analysis that would show no differences between the two groups in several factors known to cause osteoporosis is needed. Second, there was no BMD result measured with DXA in the same patient. If we compared DXA and QCT BMD measurements directly in the same patients to identify differences in measurement methods, then our theory could be more clearly confirmed. However, the lack of data using DXA in the study forced a comparison with data from previous studies. Lastly, the number of patients with AS was relatively

small. Our findings need to be confirmed in other large populations of AS patients in the future.

On the basis of results of this study, we propose several methods for the prevention and treatment of osteoporosis in patients with AS in the future. First, QCT should be performed instead of DXA in patients with advanced ankylosis. If osteopenia or osteoporosis is identified, treatment with vitamin D or osteoporosis agents should be considered. BMD measurements using QCT may detect bone loss earlier than measurements using DXA, allowing the early diagnosis, prevention and treatment of osteoporosis. A recent study demonstrated that low BMD predicts radiographic progression in axial spondyloarthritis (axSpA) [5]. In another study, the presence of baseline syndesmophytes in axSpA was found to most strongly predict the formation of new syndesmophytes. Similar results were found in early axSpA patients and AS patients with regard to spinal radiographic progression [13, 46, 47]. These findings support the usefulness of early identification of bone loss for preventing spinal radiographic progression, and suggest that it should be applied to patients with ankylosis as well as to patients with no syndesmophytes identified. Another important factor is to encourage patients to exercise and stretch to maintain their physical activity. Therapeutic exercise as well as medication is an important factor in AS management [48]. Previous reviews have confirmed that expert-led exercise-based interventions are effective in improving physical function, mobility, disease activity and quality-of-life outcomes [49, 50]. Despite these benefits, individuals with AS have low compliance to exercise programs, curtailing the efficacy of exercise-based interventions. Therefore, it will be necessary to understand the importance of exercise and to actively make adjustments so that the patient's pain would not be a problem hindering to exercise.

This study suggests that BMD reduction may be more common than is known in patients with AS, and it is meaningful that the analysis was conducted using QCT. Patients with bone bridges showed a reduction in BMD in QCT measurements, although it is possible that DXA measurements would yield normal BMD values in the same patients. Therefore, we believe that this finding indirectly supports the rationale for introducing QCT, not DXA, as a standard method for measuring BMD in patients with AS.

## Conclusion

In patients with AS, the bone bridge significantly affects the decrease in bone density. Our findings need to be confirmed in other large populations of AS patients in the future.

## Author Contributions

**Formal analysis:** Hyung In Yang, Yeon-Ah Lee, Seung-Jae Hong.

**Investigation:** Seong Jong Yun.

**Software:** Sang Wan Chung.

**Writing – original draft:** So Yun Lee.

**Writing – review & editing:** Ran Song, Sang-Hoon Lee.

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
