## [Decision Letter · Decision Letter 0]

11 Jan 2021

PONE-D-20-33718

The Bone Bridge Significantly Affects the Decrease in Bone Mineral Density Measured with Quantitative Computed Tomography in Ankylosing Spondylitis.

PLOS ONE

Dear Dr. Lee,

Thank you for submitting your manuscript to PLOS ONE. After careful consideration, we feel that it has merit but does not fully meet PLOS ONE’s publication criteria as it currently stands. Therefore, we invite you to submit a revised version of the manuscript that addresses the points raised during the review process.

The review panel was enthusiastic about this manuscript and had overall minor concerns about addressing non-clinical factors that may contribute to osteoporosis in AS patients.  Although the panel is in agreement that qCT is a better method, the authors also need to reinforce the reason to not pursue DXA, since it is still considered the clinical gold standard for bone mineral measurements, and the audience of the Journal may look for that link between methods. Lastly, a power analysis also is deemed necessary to lend support to the statistical findings.

We look forward to receiving your revised manuscript.

Kind regards,

Alejandro A. Espinoza Orías, PhD

Academic Editor

PLOS ONE

Journal Requirements:

2. Thank you for stating in the text of your manuscript "The Institutional Review Board (IRB) approval for the study was received, Informed consent was waived by the board. Retrospective data collection was approved by the IRB of Kyung Hee University Hospital, Gangdong (IRB No. 2019-08-012)." Please also add this information to your ethics statement in the online submission form.

3. Thank you for stating the date range when patients were diagnosed. Please also include the date(s) on which you accessed the records to obtain the retrospective data used in your study.

4.PLOS requires an ORCID iD for the corresponding author in Editorial Manager on papers submitted after December 6th, 2016. Please ensure that you have an ORCID iD and that it is validated in Editorial Manager. To do this, go to ‘Update my Information’ (in the upper left-hand corner of the main menu), and click on the Fetch/Validate link next to the ORCID field. This will take you to the ORCID site and allow you to create a new iD or authenticate a pre-existing iD in Editorial Manager. Please see the following video for instructions on linking an ORCID iD to your Editorial Manager account: https://www.youtube.com/watch?v=_xcclfuvtxQ

5.Thank you for submitting the above manuscript to PLOS ONE. During our internal evaluation of the manuscript, we found some minor occurrences of overlapping text with the following previous publication(s), some of which you are an author, which needs to be addressed:

- https://arthritis-research.biomedcentral.com/articles/10.1186/s13075-018-1731-8

- https://www.ncbi.nlm.nih.gov/pmc/articles/PMC3623474/

We would like to make you aware that copying extracts from previous publications word-for-word, especially outside the methods section, is unacceptable. In addition, the reproduction of text from published reports has implications for the copyright that may apply to the publications.

Please revise the manuscript to quote or rephrase the duplicated text and cite your sources for text outside the methods section. Please note that further consideration is dependent on the submission of a manuscript that addresses these concerns about the overlap in text with published work.

<h1>** **</h1>

Reviewers' comments:

Reviewer's Responses to Questions

**Comments to the Author**

1. Is the manuscript technically sound, and do the data support the conclusions?

Reviewer #1: Yes

Reviewer #2: Yes

2. Has the statistical analysis been performed appropriately and rigorously? 

Reviewer #1: Yes

Reviewer #2: Yes

3. Have the authors made all data underlying the findings in their manuscript fully available?

Reviewer #1: Yes

Reviewer #2: Yes

4. Is the manuscript presented in an intelligible fashion and written in standard English?

Reviewer #1: Yes

Reviewer #2: Yes

5. Review Comments to the Author

Reviewer #1: This is a well written article examining the association between bone bridging, lumbar ROM and QCT assessed BMD. Using retrospective data the authors are able to effectively demonstrate a negative correlation between QCT BMD, bone bridging, and lumbar ROM. The discussion provides a good frame work for the utilization of this association. Osteoporosis in AS patients is likely multifactorial based on the chronic inflammatory state as well as decreasing physical activity (lumbar ROM).

The limitation of this article is the relatively small data set. For example, although statistically insignificant in their evaluation, the use of TNF inhibitors differed by about 20% between the normal and low BMD groups, begging the question, would these medications have some significance in a larger data set.

This article is also limited in terms of the use of QCT for the evaluation of BMD in this patient population. Although QCT appears to correlate with bone bridging and lumbar ROM, without a DXA value to compare to as an internal control (from both the hip and lumbar spine), I don’t think the rationale for utilization of QCT over DXA is entirely supported.

Overall, this is an interesting article that introduces the interplay between bone bridging, physical activity (ROM), and osteoporosis in the AS population. Although it does not provide any detail of causality in regards to this interplay, this introduces a new avenue for future research on this topic.

Reviewer #2: The purpose of this study was to determine whether the formation of a bone bridge correlates with low BMD measured

with QCT in patients with AS.

Authors concluded that the bone bridge significantly affects the decrease in bone density in patients with AS.

This study could support the usefulness of early identification of bone loss to prevent spinal radiographic progression of AS.

However, pathogenesis of osteoporosis could be multifactorial and bony bridge might not be the main pathology of osteoporosis in AS patients. There could be possibilities that osteoporosis is more related to socioeconomic status, occupational status, and disability caused by spinal malalignment. Author should decribe this.

6. PLOS authors have the option to publish the peer review history of their article (what does this mean?). If published, this will include your full peer review and any attached files.

Reviewer #1: No

Reviewer #2: No

---

## [Author Response · Author response to Decision Letter 0]

2 Mar 2021

To the Editor-in-Chief, PLOS ONE

Dear Editor:

First of all, thank you very much for your consideration of publication of this paper in PLOS ONE. We made efforts to follow your recommendations and suggestions as best as we could. We addressed several important concerns that the reviewers brought up and tried to revise the manuscript appropriately. We hope that our response and revision can alleviate the reviewers’ concerns. 

Thank you once again for your time and consideration.

Sincerely Yours,

Sang-Hoon Lee, MD

The review panel was enthusiastic about this manuscript and had overall minor concerns about addressing non-clinical factors that may contribute to osteoporosis in AS patients. Although the panel is in agreement that qCT is a better method, the authors also need to reinforce the reason to not pursue DXA, since it is still considered the clinical gold standard for bone mineral measurements, and the audience of the Journal may look for that link between methods. Lastly, a power analysis also is deemed necessary to lend support to the statistical findings.

> Thank you for your thoughtful comment. We have mentioned whether DXA use is not appropriate in AS patients. In AS, it is known that BMD is overestimated due to ossification of ligament around the vertebral body (reference #23). In the previous study, when using DXA to diagnose osteoporosis, fewer patients were diagnosed with osteoporosis than when measured by QCT. Diagnosed (reference #28). Although DXA is still the most used diagnostic method for osteoporosis, it should be considered as the diagnosis through DXA may not be as accurate as in AS. I have marked the sentences related to the first page of the discussion. 

We also performed a power analysis in response to your comment. As a result of post hoc analysis on the 47 patients used in this study, the power was 0.96 at ρ0 = -0.501, α = 0.05. This is added at the end of the 'Correlation between bone bridge and QCT BMD' part of the result.

Reviewers' comments to the author:

Reviewer #1: 

This is a well written article examining the association between bone bridging, lumbar ROM and QCT assessed BMD. Using retrospective data the authors are able to effectively demonstrate a negative correlation between QCT BMD, bone bridging, and lumbar ROM. The discussion provides a good frame work for the utilization of this association. Osteoporosis in AS patients is likely multifactorial based on the chronic inflammatory state as well as decreasing physical activity (lumbar ROM).

The limitation of this article is the relatively small data set. For example, although statistically insignificant in their evaluation, the use of TNF inhibitors differed by about 20% between the normal and low BMD groups, begging the question, would these medications have some significance in a larger data set.

> Thank you for your thoughtful comment. We agree with your point. As there have been previous studies showing that anti-TNF therapy slows down generalized bone loss in AS patients, an accurate analysis is essential that there is no difference in anti-TNF therapy between the two groups when the sample size increases. The content of the positive effect of the TNF inhibitor was mentioned in the text.

This article is also limited in terms of the use of QCT for the evaluation of BMD in this patient population. Although QCT appears to correlate with bone bridging and lumbar ROM, without a DXA value to compare to as an internal control (from both the hip and lumbar spine), I don’t think the rationale for utilization of QCT over DXA is entirely supported.

> Thank you for your thoughtful comment. As mentioned in the text, if there was a direct comparable DXA value, it would have been a sufficient basis for performing QCT rather than DXA in AS patients. However, it is difficult to use both methods when measuring BMD in one patient due to problems such as insurance and cost of DXA test. In the study we cited as reference # 28, it was confirmed that BMD is overestimated in DXA when compared to QCT in patients with AS due to spinal involvement. In this study, it was mentioned in a limitation that the analysis of BMD was not performed in both DXA and QCT methods. 

Overall, this is an interesting article that introduces the interplay between bone bridging, physical activity (ROM), and osteoporosis in the AS population. Although it does not provide any detail of causality in regards to this interplay, this introduces a new avenue for future research on this topic.

Reviewer #2: 

The purpose of this study was to determine whether the formation of a bone bridge correlates with low BMD measured with QCT in patients with AS.

Authors concluded that the bone bridge significantly affects the decrease in bone density in patients with AS.

This study could support the usefulness of early identification of bone loss to prevent spinal radiographic progression of AS.

However, pathogenesis of osteoporosis could be multifactorial and bony bridge might not be the main pathology of osteoporosis in AS patients. There could be possibilities that osteoporosis is more related to socioeconomic status, occupational status, and disability caused by spinal malalignment. Author should describe this.

> Thank you for your thoughtful comment. We agree with your point. In the baseline characteristics analysis, one of our limitations is that the factors known to cause osteoporosis were not sufficiently identified. In response to this comment, we emphasized by adding content to the text.

---

## [Editor Report · Decision Letter 1]

22 Mar 2021

The Bone Bridge Significantly Affects the Decrease in Bone Mineral Density Measured with Quantitative Computed Tomography in Ankylosing Spondylitis.

PONE-D-20-33718R1

Dear Dr. Lee,

We’re pleased to inform you that your manuscript has been judged scientifically suitable for publication and will be formally accepted for publication once it meets all outstanding technical requirements.

Kind regards,

Alejandro A. Espinoza Orías, PhD

Academic Editor

PLOS ONE
---

## [Editor Report · Acceptance letter]

8 Apr 2021

PONE-D-20-33718R1 

The bone bridge significantly affects the decrease in bone mineral density measured with quantitative computed tomography in ankylosing spondylitis. 

Dear Dr. Lee:

I'm pleased to inform you that your manuscript has been deemed suitable for publication in PLOS ONE. Congratulations! Your manuscript is now with our production department. 

Kind regards, 

on behalf of

Dr. Alejandro A. Espinoza Orías 

Academic Editor

PLOS ONE